# Cytoskeleton polarity is essential in determining orientational order in basal bodies of multi-ciliated cells

Toshinori Namba[1], Shuji Ishihara[1,2]*

**1** Graduate School of Arts and Sciences, The University of Tokyo, Komaba, Tokyo, Japan, **2** Universal Biology Institute, The University of Tokyo, Komaba, Tokyo, Japan

* csishihara@g.ecc.u-tokyo.ac.jp.

**Data Availability Statement:** All relevant data are within the manuscript and its Supporting Information files.

**Funding:** This research was supported partly by grants from the Core Research for Evolutional

## Abstract

In multi-ciliated cells, directed and synchronous ciliary beating in the apical membrane occurs through appropriate configuration of basal bodies (BBs, roots of cilia). Although it has been experimentally shown that the position and orientation of BBs are coordinated by apical cytoskeletons (CSKs), such as microtubules (MTs), and planar cell polarity (PCP), the underlying mechanism for achieving the patterning of BBs is not yet understood. In this study, we propose that polarity in bundles of apical MTs play a crucial role in the patterning of BBs. First, the necessity of the polarity was discussed by theoretical consideration on the symmetry of the system. The existence of the polarity was investigated by measuring relative angles between the MTs and BBs using published experimental data. Next, a mathematical model for BB patterning was derived by combining the polarity and self-organizational ability of CSKs. In the model, BBs were treated as finite-size particles in the medium of CSKs and excluded volume effects between BBs and CSKs were taken into account. The model reproduces the various experimental observations, including normal and drug-treated phenotypes. Our model with polarity provides a coherent and testable mechanism for apical BB pattern formation. We have also discussed the implication of our study on cell chirality.

## Author summary

Synchronous and directed ciliary beating in trachea allows transport and ejection of virus and dust from the body. This ciliary function depends on the coordinated configuration of basal bodies (root of cilia) in apical cell membrane. However, the mechanism for their formation remains unknown. In this study, we show that the polarity in apical microtubule bundles plays a significant role in the organization of basal bodies. A mathematical model incorporating polarity has been formulated which provides a coherent explanation and is able to reproduce experimental observations. We have clarified both necessity ('why polarity is required for pattern formation') and sufficiency ('how polarity works for pattern formation') of cytoskeleton polarity for correct pattering of basal bodies with verification by experimental data. This model further leads us to a possible mechanism for cellular chirality.

Science and Technology of the Japan Science and Technology Agency (to SI) and in part by the Japan Society for the Promotion of Science Grants-in-Aid for Scientific Research in Innovative Areas (25103008 and 12010883 to SI). The funders had no role in study design, data collection and analysis, decision to publish, or preparation of the manuscript.

## Introduction

Synchronous and coordinated ciliary beating underlies various biological functions in organisms ranging from ciliates to mammals [1]. In the mammalian trachea, synchronous ciliary beating of multi-ciliated cells produces mucociliary transport, that is the flow of mucus to the oral side of the epithelial cell surface, ejecting viruses and dust to the outside of the body [2]. For efficient mucociliary transport, hundreds of cilia covering the apical membrane of a cell beat synchronously in the same direction. This organized directionality in ciliary beating is achieved by basal bodies (BBs), which are roots of the cilia. These BBs point in the same orientation (Fig 1A), which is defined by the relative position of the basal foot (BF), a major appendage associated with BBs. In addition, previous studies have revealed that BBs are regularly aligned in the apical membrane of the mature epithelial cells in mouse trachea [3]. The

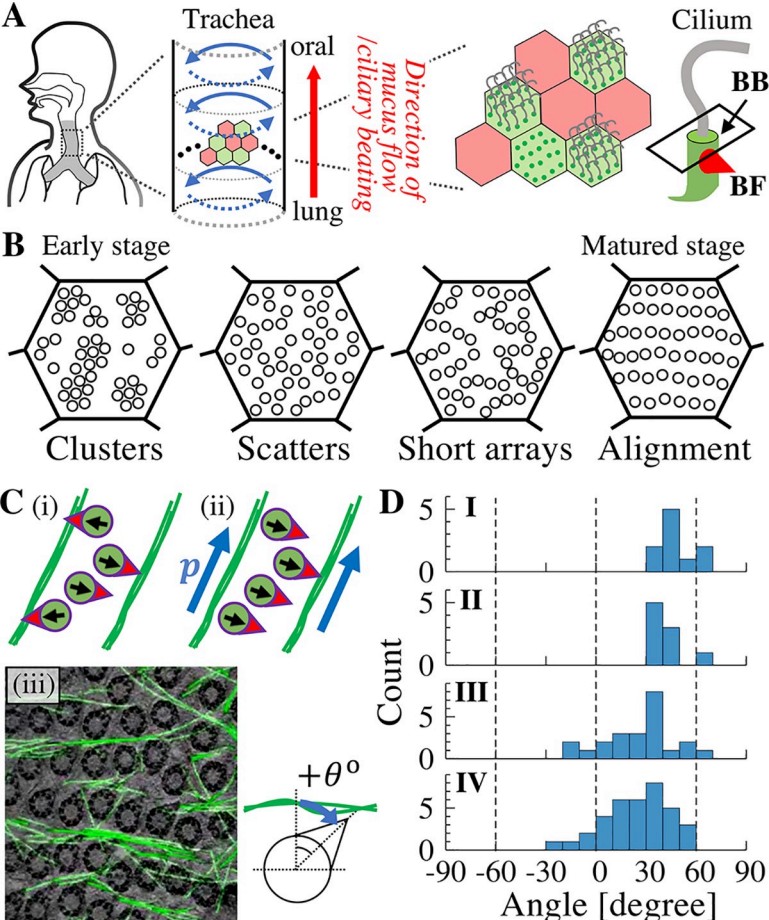

**Fig 1. Schematics of the positional and orientational orders of BBs in the trachea.** (A) The mammalian trachea sends mucus to the mouth on the surface of tracheal epithelial cells (red arrow). Tracheal epithelia is composed of multi-ciliated (green) and mucous cells (red). The mucus flow is produced by beating of multiple cilia in the apical membrane of the multi-ciliated cells. The orientation and positioning of cilia are determined by BF (red corn) and BBs (green rod), respectively. BF is an MT-organizing center [8,9]. (B) Typical pattern of BB distribution. During maturation of multi-ciliated cells, the pattern of BBs is changed from cluster to alignment patterns [3]. (C) Polar MTs can orient BBs to the definite direction. (i) Binding of the BF to the neighboring MT bundles is not sufficient to realize unidirectional orientation of BBs [10]. (ii) BBs can orient to same direction if polarity exists in the MT bundles (denoted as $p$) and BBs can recognize it. The polarity is also shown by blue arrows in panel A. (iii) TEM images of BBs and detected MTs (adapted from [3]). BBs contact MTs at a biased angle. (D) Histograms of the angle of BFs measured using TEM (From (I) Movie S8 in [11], (II) Fig 6 in [11] (III) Video 6 in [3], and (IV) Video 7 in [3]).

positional and orientational order of BBs is crucial for proper ciliary beating, as reported in many other systems where disturbance in ciliary organization resulted in dysfunction of organs, such as inner ear hearing loss [4], infertility [5], hydrocephalus [6], and situs inversus [7]. Therefore, elucidation of the mechanism of ciliary organization on the apical membrane is of paramount biological and clinical importance. However, it is largely unknown how BBs are coordinated through the maturation of the tracheal multi-ciliated cells.

Long-term live cell imaging of cultured tracheal cells revealed that maturation of BBs in the apical membrane proceeds though several stages, characterized by a typical pattern in BB distribution (Fig 1B) [3]. Clusters of BBs are observed in the early stages after differentiation to multi-ciliated cells. The clusters are then scattered, followed by formation of short and branched arrays of several BBs termed as 'partial alignment' [3]. Through connection and reconnection of these arrays, BBs finally become regularly aligned over the apical membrane. The number of BBs remains nearly unchanged during the maturation period. BBs become increasingly oriented in the same direction indicated by the relative position of the BF, concurrent with the alignment of BBs. Quantification of positional and orientational orders of BBs showed that these two ordering processes in position and orientation were highly correlated [3], suggesting that both are coordinated by common factors. Cytoskeletons (CSK) in the apical membrane were shown to be involved in the regulation of BB distribution. During maturation of cells, microtubules (MTs) and 10 nm-filaments, which are sparsely distributed in the early stage, grow to a dense mesh network and surround the BBs [12]. Most MT bundles run in parallel between the aligned BB array, which might contribute to the positioning formation of the BB merely by invading and separating the array of BBs [11]. Further, since BF directly binds to MTs as confirmed by transmission electron microscopy (TEM) for which γ-tubulin in BF might mediate the association [9], it was suggested that the orientation of the BBs is also regulated by MTs [11]. Remarkably, inhibition of MT polymerization by nocodazole treatment results in disruption of BB alignment pattern and formation of BB cluster pattern, with disruption in the orientation of BB as well. Removal of nocodazole by washing results in recovery of BB positional order [3]. In summary, there is sufficient experimental evidence to show that CSKs are responsible for the determination of position as well as orientation of BBs. However, the underlying mechanism explaining these observations remains elusive. In particular, the attachment of BF to MTs might be important for the determination of BB orientation [9], but its mechanistic details have not been identified as yet.

In the present study, we have generated a mathematical model to explain the ordering of position and orientation in BBs. Our aim here is not to propose a detailed model that can explain all aspects of the experimental findings, but rather to provide a possible model to explore the role of MTs for the observed BB patterning. Firstly, the symmetry of the system has been taken into consideration to show that determination of orientation in BBs requires polarity in MT bundles. The presence of polarity is supported by a quantitative relationship between BB orientation and CSK bundles, determined from available experimental data [3,11]. This was followed by derivation of a mathematical model based on the characteristics of CSKs and their interaction with BBs. 'Active matter' formalism was employed by which the generic form of system equations is obtained by taking into account the symmetry of the system [13,14]. This kind of formulation was conducted for studying pattern formation on apical membranes for a system without BBs [15]. Our early study on tracheal cells also employed this scheme and produced positional order of BBs [3], but did not consider the orientation of BBs and the CSK-BB interaction. Using the derived model, we have investigated the conditions for BBs to align and point in the same direction, which further confirms that MT polarity is crucial for the establishment of BB pattern. The information transmission from planar cell polarity (PCP) to the BB orientation has also been discussed. In the Discussion, we have summarized our

findings and challenges that need to be addressed in future. We have also considered the implication of our findings in determining a possible mechanism for cellular chirality in determining the relationship between PCP and CSK polarity.

## Results

### Polarity in cytoskeleton guides orientation of BBs

During the maturation period of multi-ciliated cells, BBs establish regularity in their position and orientation. The patterning of BB arrays follows the formation of the CSK matrix, particularly the MT bundles forming on the cytosolic apical membrane [3,11]. The BBs orient in the same direction at the same time [3]. Since the BF, which is an appendage of BBs, has a component binding to MTs as confirmed by TEM images and mutant strains in Odf2 [8,9,11], it was suggested that the connection of BF with neighboring MT bundles is a determinant of BB orientation [10]. However, this binding between BF and MTs allows BBs to associate with either of the two neighboring MT bundles and hence they cannot distinguish between the two opposite directions (Fig 1C (i)). Therefore, how BBs can determine their definite direction remains a mystery.

A probable hypothesis to fill this knowledge gap is the presence of polarity in the MT bundles. It may be assumed that there is a component in CSK that provides the direction which the BBs are able to interpret. The presence of polarity therefore enables BBs to distinguish the direction by breaking the symmetry in the relationship between BBs and MT bundles (Fig 1C (ii)). To assess this hypothesis, we focused on the contact angles between BBs and MT bundles, since the angles are apparently neither perpendicular nor uniformly distributed but are specifically biased. We measured the angles between MTs and the orientation of the associated BBs from published TEM images (Fig 1C (iii)) [3,11]. The contact angles were biased to the positive side as shown in Fig 1D and Table 1 (n = 4). This observation indicates a polar nature in MTs and chirality in BF, which enables the connection of BBs to MTs within a specific range of angles.

### Mathematical model for the interaction between CSK and BB

A mathematical model was generated based on the polarity of MTs, demonstrating that polarity is a necessary and sufficient condition for the reproduction of the observed regular positional and orientational orders of BBs on the apical membrane. The following processes were taken into account while generating the model (Fig 2);

i. CSKs form bundles on the apical membrane that sustain the intrinsic polarity.

ii. CSKs and BBs exclude each other.

iii. BBs repel each other.

**Table 1. Measured angles of BFs and its numbers in Fig 1D.**

| TEM images | mean | S.D. | numbers | Reference |
|:---:|:---:|:---:|:---:|:---:|
| I | 47.2˚ | ±10.5˚ | 10 | Movie S8 in [11] |
| II | 41.2˚ | ±8.6˚ | 9 | Fig 6 in [11] |
| III | 26.7˚ | ±20.6˚ | 23 | Video 6 in [3] |
| IV | 24.3˚ | ±19.0˚ | 36 | Video 7 in [3] |

Mean and standard deviation of contact angle of BBs to MTs. Each sample is adopted from TEM images shown in Reference. Mean and standard deviation are defined according to directional statistics [16].

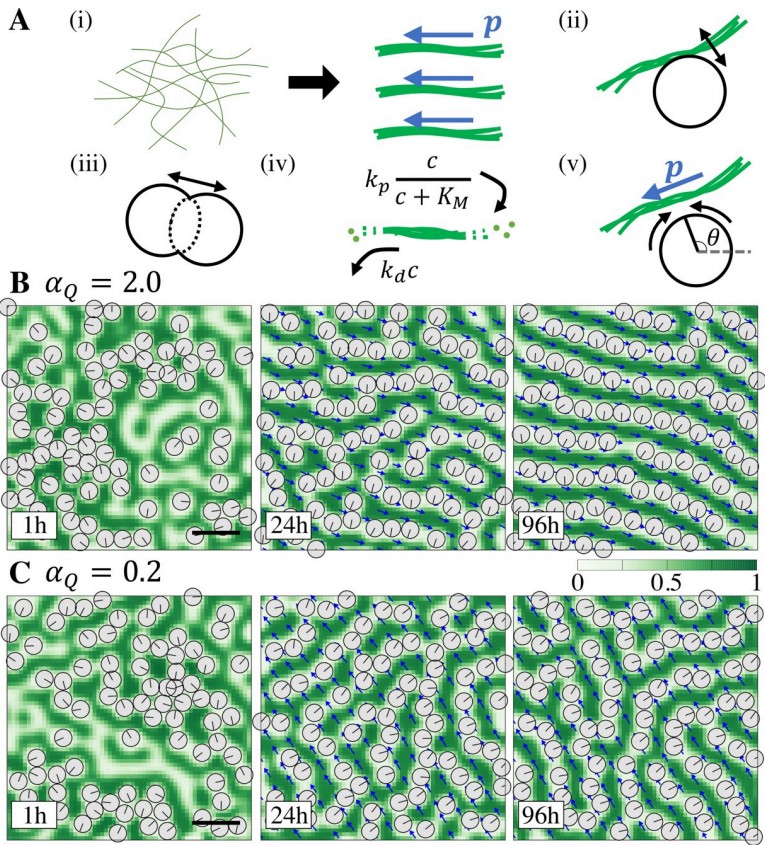

**Fig 2. Model for positional and orientational orders of BBs.** (A) Illustration of the processes considered in the model. (i) CSK filaments are condensed and bundled in parallel and their polarity orient in the same direction (blue arrow, **p**). (ii, iii) One BB repels the CSK bundle and other BBs. (iv) Polymerization and depolymerization of CSKs. (v) The orientation of BB, $\theta$, is determined by CSK polarity. (B) Typical time series of BB pattern development ($\alpha_Q$ = 2.0). Green indicates CSK concentration, and blue arrow indicates the polarity of CSK. Gray disk is a BB and the line segment in the disk indicates the orientation of BB. Patten of BBs reaches "alignment". (C) Time series when the coupling between polarity and CSK concentration is weak ($\alpha_Q$ = 0.2). Eventual pattern is "partial alignment". Bars, 0.5 μm.

iv.  Polymerization and depolymerization of CSKs.

v.  BBs orient in the direction determined by CSK polarity.

The processes (i)-(iii) are modeled by a free energy function *F*, a generic form of which is as follows [13–15,17].

$$F = F_{csk} + F_{c\phi} + F_{\phi\phi} \tag{1}$$

$$F_{csk} = \int \left[ f_c(c) + \frac{D_0}{2} [\nabla c]^t (I + \alpha_Q Q) \nabla c - \frac{\beta_P(c)}{2} |\boldsymbol{p}|^2 + \frac{1}{4} |\boldsymbol{p}|^4 + \frac{K}{2} |\nabla \boldsymbol{p}|^2 \right] dr \tag{2}$$

$$F_{c\phi} = \lambda_{c\phi} \sum_i \int \phi_i c^2 dr \tag{3}$$

$$F_{\phi\phi} = \lambda_{\phi\phi} \sum_{[i,j]} \int \phi_i \phi_j dr \tag{4}$$

Here, $c(t, \boldsymbol{x})$ and $\boldsymbol{p}(t, \boldsymbol{x})$ are continuum field variables representing concentration and polarity of CSKs at time $t$ in a two-dimensional space $\boldsymbol{x}$. $F_{csk}$ is the free energy of bundling and polarity of CSK. We employed a simple Landau energy form $f_c(c) = -(\alpha_2/2)(c-c_0)^2 + (\alpha_4/4)(c-c_0)^4$, according to which, MT accumulation is induced to form bundles. In the second term, $I$ is the identity matrix and $Q = \boldsymbol{p} \otimes \boldsymbol{p} - (1/2)\mathrm{Tr}[\boldsymbol{p} \otimes \boldsymbol{p}]I$ is a nematic tensor depending on the polarity of CSKs. This term represents 'surface tension' of the MT bundles, in which, polarity $\boldsymbol{p}$ prefers parallel alignment to the CSK concentration boundary with a positive value of $\alpha_Q$, i.e., $\boldsymbol{p}$ is perpendicular to the cell concentration gradient $\nabla c$ (Fig 2A(i)). The term $F_{csk}$ describes the formation of CSK polarity. By setting $\beta_P(c) = \beta(c-c_c)$, the polarity increases with MT concentration exceeding the critical value $c_c$, up to an amplitude of $|\boldsymbol{p}| = \sqrt{\beta_P(c)}$. The last term represents the tendency of the polarity to align with each other, where one constant approximation is employed. Taken together, $F_{csk}$ provides the free energy that expresses process (i), where MT concentration and polarity are coupled via nematic tensor $Q$. The simulation results without BBs has been shown in the methods section and S1 Fig.

$F_{c\phi}$ (Eq 3) and $F_{\phi\phi}$ (Eq 4) are the free energies representing processes (ii) and (iii), respectively. The variable $\phi_i(t, \boldsymbol{x})$ represents the region occupied by the $i$-th BB, a circular disk with radius $R$ centered at $X_i$. We set $\phi_i = [1 + \tanh(\nu(R - |\boldsymbol{r} - X_i|))]/2$ by which $\phi_i = 1$ for the internal region of the BB, and $\phi_i = 0$ otherwise [18]. $\nu$ controls the steepness of the boundary of the BB surface and is set to be sufficiently large. $F_{c\phi}$ is the energy term for the excluded volume effect between BBs and CSKs, which was not considered in the previous model [3]. $F_{\phi\phi}$ is the energy term representing the excluded volume effect among BBs, to avoid overlap between different BBs, summed over all pairs of BBs.

The dynamics of the system is derived from the free energy $F$ as follows.

$$\partial_t c = \alpha_J \nabla^2 \left( \frac{\delta F}{\delta c} \right) + k_P \frac{c}{c + K_M} - k_d c \tag{5}$$

$$\partial_t \boldsymbol{p} = -\gamma_P \frac{\delta F}{\delta \boldsymbol{p}} \tag{6}$$

$$\partial_t \boldsymbol{X_i} = -\gamma_{BB} \frac{\partial F}{\partial \boldsymbol{X_i}} + \xi_i \tag{7}$$

In Eq (5), time evolution of CSK concentration $c$ is derived from the energy relaxation with mass conservation, where $\alpha_J$ is the kinetic coefficient. Polymerization and depolymerization (process iv) are represented by $k_p$ and $k_d$ being maximum polymerization rate and depolymerization rate, respectively, with $K_M$ being a constant. The mean concentration of CSKs in the system is determined by the balance between these rates. We assumed $k_p$ is rescaled by $k_d$ to keep the MT concentration almost constant (see Table 2). The time evolution rule of polarity, $\boldsymbol{p}$, and the center of the $i$-th BB, $\boldsymbol{X}_i$, are determined by simple relaxation of the free energy given by the derivative of $F$. In Eq (7), the center of the $i$-th BB moves by repulsive interaction with CSKs and other BBs. Stochastic noise is added to the evolution of $\boldsymbol{X}_i$, where $\xi_i$ is a white Gaussian random variable obtained from the normal distribution with zero-mean and standard deviation $\sigma_{BB}\Delta t$ at each time step, where $\Delta t$ is the discretized time step in the simulation. Finally, we assume that the polarity of CSKs guides BB orientation via the interaction between CSKs and BFs (process v). Although there is a preferred angle between the direction of CSK polarity and orientation of a BB (Fig 2A(v)), the preferred angle has been assumed to be $\pi/2$ in

**Table 2. Parameters used in Fig 2B.**

| Parameters | | |
|---|---|---|
| **Free energy** | | |
| $\alpha_2$ | 1.0 | Constant of 2nd order term in Landau expansion |
| $\alpha_4$ | 8.0 | Constant of 4th order term in Landau expansion |
| $c_0$ | 0.5 | Concentration on a steady state |
| $D_0$ | 0.5 | Diffusion constant |
| $\alpha_Q$ | 2.0 | Anisotoropic diffusion constant along $\boldsymbol{p}$ |
| $\beta$ | 1.5 | Contribution from concentration to polarity (Active term) |
| $c_c$ | 1/3 | Threshold of CSK concentration at which polarity can grow |
| $K$ | 3.0 | Bending modulus of CSK |
| $\lambda_{c\phi}$ | 1.0 | Repressive effect between BB and CSK |
| $\lambda_{\phi\phi}$ | 1.0 | Repressive effect between BBs |
| $\nu$ | 10.0 | Steepness of the edge of BB surface |
| $R$ | 2.5 | Radius of the disk-shaped BB |
| **Dynamics** | | |
| $\alpha_J$ | 1.5 | Constant kinetic coefficient |
| $k_p$ | $k_d(c_0+K_M)$ | Rate of polymerization of CSK* |
| $k_d$ | $\sqrt{0.1}$ | Rate of depolymerization of CSK |
| $K_M$ | 0.2 | Michaelis Menten constant in the polymerization of CSK |
| $\gamma_P$ | 0.5 | Inverse time constant for the changing of polarity |
| $\gamma_{BB}$ | 0.3 | Inverse time constant for the movement of BB |
| $\gamma_{BF}$ | 0.5 | Inverse time constant for the orientation of BF |
| **Noise strength** | | |
| $\sigma_{BB}$ | 0.2 | Standard deviation of white Gaussian random variable $\xi_i$ |
| $\sigma_{BF}$ | 0.05 | Standard deviation of white Gaussian random variable $\xi_{BF}$ |
| **System Parameters** | | |
| N | 80/20/30 | Numbers of BBs (Fig 2, Fig 4A, 4D and 4E / Fig 3 and Fig 4B / Fig 5) |
| $L_x$ | 64/32/30 | Length of $x$-axis (same as above) |
| $L_y$ | 64/32/45 | Length of $y$-axis (same as above) |
| dx | 1.0 | Spacing of square grid |

* $k_p$ is kept to be proportional to $k_d$ for the MT concentration to be almost constant.

this model. This assumption does not change the generality of the model.

$$\partial_t \theta_i = -\gamma_{BF}(\langle c\boldsymbol{p}\rangle_{\phi_i}) \cdot \boldsymbol{n}_i + \xi_{BF} \tag{8}$$

Here, $\theta_i$ is the orientation of the $i$-th BB and $\boldsymbol{n_i} = (\cos\theta_i, \sin\theta_i)$ is its director. The angle brackets $\langle \cdots \rangle$ denote the average over the surface of the BB. Gaussian white noise term $\xi_{BF}$ is also added with standard deviation $\sigma_{BF}\Delta t$.

The above argument provides the closed form of the equations to be explored below. We have numerically solved non-dimensionalized equations where the units of time and length in the simulations correspond to 5 min and 0.04 $\mu$m, respectively. Details of the equations, parameter values, and numerical methods have been provided in the methods section.

## The mathematical model reproduces BB positioning and orientation

Depending on the parameter values used in the model equations, several patterns were observed in the numerical simulations. Fig 2B shows an example of the time series obtained in

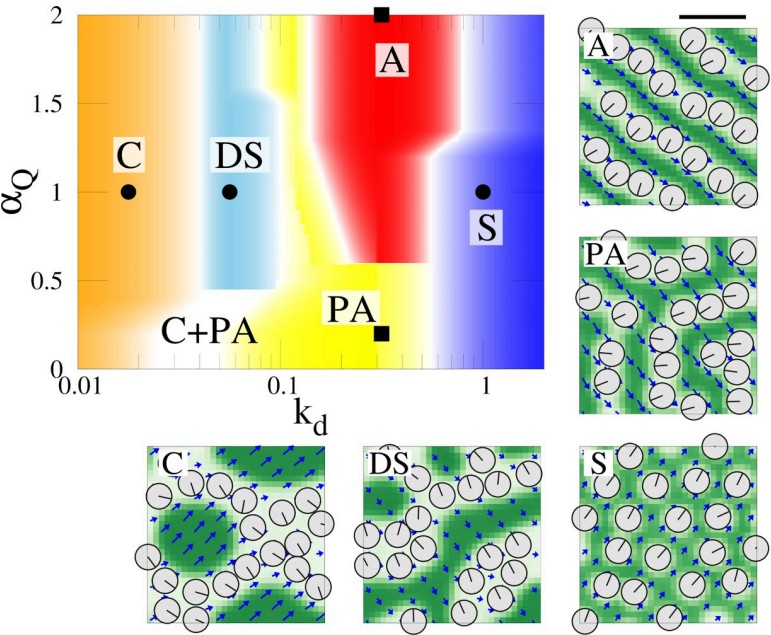

**Fig 3. Phase diagram of BB pattern against parameters $\alpha_Q$ and $k_d$.** $\alpha_Q$ is the coupling constant between CSK concentration and polarity. $k_d$ is the polymerization/depolymerization rate of CSK. Patterns are classified into Alignment (A), Cluster (C), Scatter (S), partial alignment (PA), and Double strand (DS). Typical BBs pattern in each region is shown are also shown. Bars, 0.5 μm.

the simulation when $\alpha_Q$, the coupling strength between the polarity and concentration of CSK in Eq (2), is set at $\alpha_Q$ = 2.0 (see Table 2 for the other parameters). The polarity of CSK, **p,** becomes oriented in the same direction as the system and the CSK concentration, *c*, becomes striped along the direction of the polarity. BB arrays are formed and aligned between the stripes of CSK concentration, with all BBs being oriented in the same direction as the polarity. This pattern obtained in the numerical simulation recapitulates the experimental observations in normal conditions. We refer to this pattern as 'alignment'. Decreasing $\alpha_Q$ retains direction of the polarity with stripes of CSK concentration often becoming branched and BB arrays becoming winding (Fig 2C, $\alpha_Q$ = 0.2), which is referred to as 'partial alignment' [3]. This is due to the weak coupling between polarity and CSK concentration, suggesting the importance of polarity in establishing an alignment pattern. This will be further studied in the next section.

Fig 3 illustrates the phase diagram of the patterns observed by the numerical simulation against $\alpha_Q$ and the polymerization/depolymerization rate $k_d$. Some typical patterns are shown in the diagrams which are dependent on the parameter values. The boundaries between these patterns are not so sharp due to the initial condition dependence and difficulty in classification, particularly due to the appearance of mixed patterns. For weak polymerization rate ($k_d < 10^{-1.5}$), clusters of BBs appear regardless of the value of $\alpha_Q$. By increasing $\alpha_Q$, double strand arrays of BBs are obtained. Such a pattern is occasionally observed in cultured tracheal cells [3]. On the other hand, BBs are scattered for sufficiently high polymerization rate ($k_d > 10^{0.75}$) since the amounts of CSKs surrounding BBs recover quickly to the mean concentration and isolate them. Alignment and partial alignment patterns appear in the middle region of $k_d$, depending on the value of $\alpha_Q$ as mentioned above. Altogether, configuration of BBs is dependent on the polymerization rate and coupling strength between CSK polarity and concentration. Appearance of BB clusters at low polymerization rate is of interest since it resembles those observed in the nocodazole treated clusters [3], which will be discussed further.

## Polarity is required for alignment pattern of BBs

To further explore the role of polarity in CSKs, we studied the change in the behavior of the system with change in the parameters $\alpha_Q$ and $K$, the latter is bending modulus of MT bundles (Eq 2). Our numerical simulation suggests that these two parameters must be sufficiently high for achieving regular alignment pattern of BBs, indicating importance of the polarity in CSKs for BB alignment pattern. In S2 Fig, the straightness of BB arrays is quantified by calculating $\langle -\cos\psi_i \rangle$, which is the average of the negative cosine of the opening angle between line segments connecting the $i$-th BB and its two neighboring BBs. When either $\alpha_Q$ and $K$ are low, arrays of BBs are short or are not aligned straight, classifying them into 'partial alignment' or a 'scatter pattern'. Therefore, polarity in CSKs plays two roles in our model. It does not only convey the orientational information to BBs (Eq 8), but also enables BBs to be regularly aligned through the interaction with CSK concentration and BBs.

## Bundling of CSKs enhances BB alignment formation

MTs are bundled in the apical membrane, which stiffen the apical CSK network and could contribute towards the formation of aligned BB arrays. For more insights into the role of the MT bundles for BB pattern, we changed the functional form of the free energy $f_c(c)$ in Eq (2) which drives the formation of MT bundles (condensation of MTs). We re-parametrized the free energy as $f_c(c) = \lambda_W(c-c_0-W)^2(c-c_0+W)^2$ in which the parameter $\lambda_W$ controls the scale of the energy (i.e., 'depth' of the potential), while the positions of the two minima are controlled by $W$. With a proper form of $f_c(c)$, BB arrays are well lined up as shown in S3 Fig, consistent with the preceding sections. Since in the system without BBs, the double minimum form is indispensable for inducing the phase separation (condensation) of CSKs resulting in the stripe pattern (see Methods and S1 Fig), we hypothesize $W$ to be positive-finite. If that were not the case, BBs would scatter and be randomly configured. This is confirmed for large $\lambda_W$ and small $W$ where BBs fails to align (S3(iv) Fig). However, when both $\lambda_W$ and $W$ are small, there were unexpected observations in which arrays of BBs were formed and were aligned (S3(ii) Fig). Since the form of $f_c$ is flattened in the parameter region, we then conducted numerical simulation by erasing the free energy (i.e., $f_c(c) = 0$). We found that the aligned pattern of BBs is achieved as shown in Fig 4A. The obtained phase diagram on $k_d$ - $\alpha_Q$ plane shown in Fig 4B is similar to those obtained in Fig 3, indicating that the condensation of CSKs is not essential for the pattern of BBs. However, the region of the 'aligned pattern' becomes narrower, and the BB arrays are often branched in the aligned region (Fig 4B(i)) and are less straight. As shown in Fig 4C, in the model including $f_c(c)$ term, BBs align in straighter over a wide range of noise strengths in Eq (7). Thus, MT bundles are useful for more robust and ordered positioning of BBs.

We conclude from the above numerical results that even without the condensation effect of MTs, aligned patterns can be realized to some extent by the exclusion volume effect between BBs and CSK. This was in contrast to the results of the previous model, where exclusive interaction between BBs and CSKs was not considered and condensation of CSKs was indispensable for patterning of BBs [3]. However, it is possible that the fluctuation is suppressed and the aligned pattern of BBs is achieved more robustly through the use of MT bundles. This result is biologically relevant since the maturation of multi-ciliated cells must progress under various external disturbances [10,19].

## Simulation for nocodazole treated cells

Given the role of polarity and condensation of CSKs investigated in the preceding sections, here we studied the formation of BB clusters in cells treated with nocodazole, which inhibits

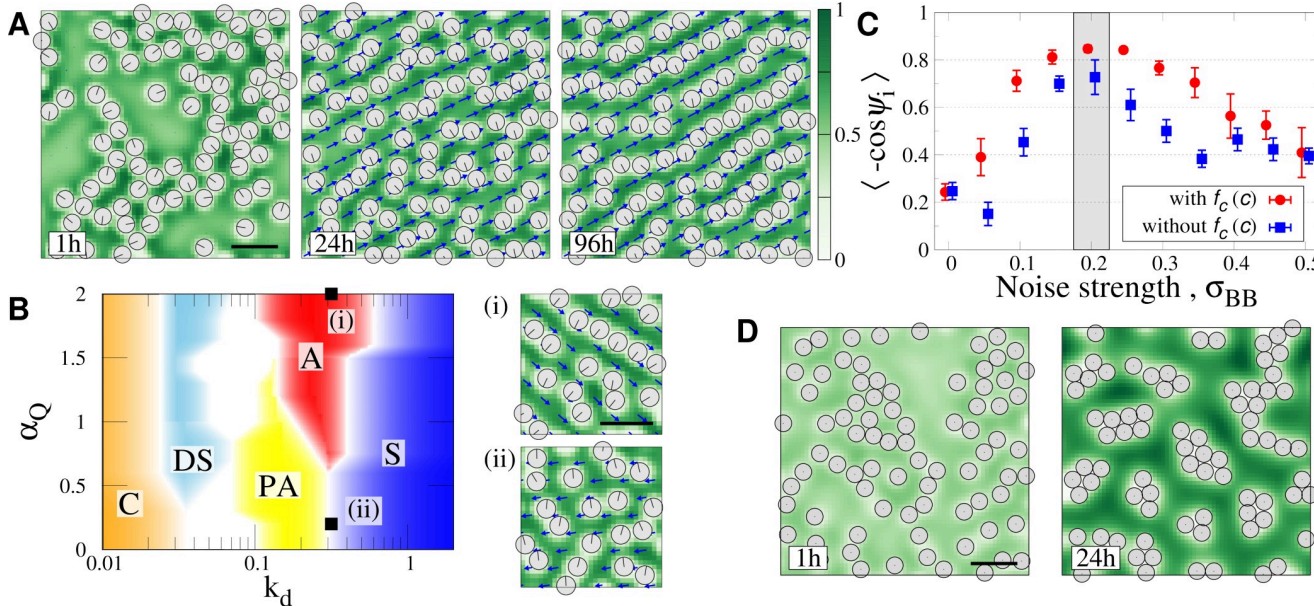

**Fig 4. Model without the condensation term and simulation for nocodazole treated cells.** (A) Typical time series of BB pattern when $f_c(c) = 0$. Parameter values are the same as Fig 2B. (B) Phase diagram of the model without condensation. (i) and (ii) show examples of "alignment" and "partial alignment", where the same parameters are used as corresponding points in Fig 3. (C) Robustness of BB for noise strength, $\sigma_{BB}$. The straightness of BB arrays, $\langle -\cos\psi_i \rangle$, was compared between models with and without $f_c(c)$ term. Gray shadow area indicates $\sigma_{BB}$ used in the other simulations. The error bars represent the standard error among four independent simulations. (D) Simulated patterns corresponding to nocodazole treated cells, which is produced by the model without condensation term and coupling with polarity (See Methods). Bars, 0.5 μm.

MT polymerization [3]. As shown in Figs 3 and 4B, BB clusters are formed at low polymerization and depolymerization ($k_d < 10^{-1.5}$), consistent with the experimental result. To clarify the mechanism of cluster formation, we simplified our model to ignore polarity and condensation of MTs, since they would play only an insignificant role when MT polymerization is inhibited. The simplified model is obtained by replacing the free energy $F_{csk}$ in Eq (2) with the following one;

$$F_{csk} = \int \frac{D_0}{2} [\nabla c]^2 dr. \tag{9}$$

The numerical simulation of the simplified model demonstrates the formation of BB clusters (Fig 4D), similar to the experimental observation in nocodazole treated cells. This result indicates that clustering of BBs occurs by merely minimizing the quadratic term of the concentration gradient, attributed to the depletion force [20]. Since the concentration gradient is localized in the boundary of BBs, accumulation of BBs is preferable for decreasing the total length of the boundary. Meanwhile, interaction length of the resulting attractive force between BBs is too short, resulting in the appearance of multiple clusters depending on the initial position of BBs.

## MTs transmit directional information in PCP to BBs

All the cilia orient to the oral side of the tracheal tissue *in vivo*, which is coordinated by PCP. In fact, directionality of BBs is reduced by knocking out the PCP-related protein Vangl1, compared with those in wild-type (WT) [21]. Importance of PCP in cilial organization is also evident in many systems [4,6,7,22]. In *Drosophila* epithelial cells, the apical MTs are aligned along the proximal-distal axis (coinciding with the PCP axis), which is involved in directed

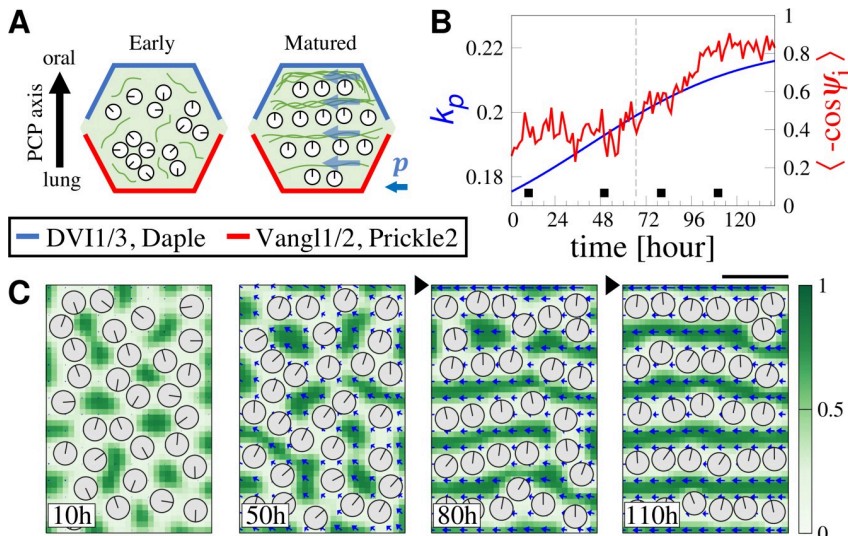

**Fig 5. Simulation of BBs positioning by PCP.** (A) PCP proteins are localized along the oral and lung boundaries on the apical membrane. CSK concentration in the apical is low in immature cell and becomes increased during maturation. In matured cells, MTs are more distributed on the oral side (indicated by localization of DVI1/3 and Daple). (B) Polymerization rate $k_p$ is increased (blue line) and the boundary condition is changed at $t = 66.7$ h (gray dashed line) in the simulation. The straightness of BB arrays obtained by simulation is shown by a red line. (C) Simulation result for pattern formation of BB alignment. Black triangle represents change in boundary condition (see text). Bars, 0.5 μm.

molecular transportation within the cell [23–25]. Recent experiments suggest that apical CSKs mediate the transmission of directional information from PCP to BBs and cilia, where PCP proteins are localized along the oral and lung side of the adherent membrane in each cell (Fig 5A). However, the detailed information transmission mechanisms remain unclear and may be different among different cell types and species [21,26].

It was observed that MT concentration is significantly high along the oral side of the cell boundary membrane in a tracheal cell, which was attributed to the interaction with PCP proteins such as Frizzled, Dishevelled, and Daple which accumulate on the oral side of the cell membrane (see Fig 5A) [3,21,27]. Although the asymmetric accumulation of the cortex MTs can also be related with MT organization over the apical membrane, and can thus proceed at the same time with the pattern formation of BBs [28], here we do not address the entire process. We restrict our study to simply demonstrate a possible mechanism of how the pattern of BBs can be modulated by the PCPs. In the framework of our model, the effect of PCP is incorporated as the boundary condition as follows [3]. For the oral side boundary, concentration of CSKs is set to be high (i.e., Dirichlet condition $c = c_H$ is applied, see Methods for details) and direction of the polarity is restricted to be parallel along the border. Neumann conditions are applied for the variables for the other boundary. In order to demonstrate the maturation process of a multi-ciliated cell during high CSK concentration, we simulated a condition where the CSK concentration on the oral boundary is switched from Neuman to Dirichlet boundary conditions mentioned above, at time $t = 66.7$ h, and the parameter $k_p$ also increases with time (Fig 5B blue line; see Methods for details).

The time series of the simulation result is shown in Fig 5C. Initially scattered pattern of BBs ($t = 10$ h in the panel) evolve to the partial alignment pattern ($t = 50$ h and 80 h), where short or branched arrays of BBs are formed by the increase in CSK concentration and BBs orient to the same direction locally (Fig 5C). BBs eventually establish a fully aligned pattern that is

parallel to the oral border through connection and reconnection of BB arrays, ($t$ = 110h). This time evolution of BB pattern agrees with those observed in the normal maturation process of a multi-ciliated cell.

## Discussion

On the apical membrane of epithelial cells, CSKs such as MTs, actins, and 10 nm-filaments can perform various biological functions through the process of pattern formation attributed to the self-organizational ability of CSKs [15,17,29–31]. In the present study, we investigated positional and orientation orders of BBs in the apical membrane of multi-ciliated cell which are responsible for coordinated ciliary beating to generate mucociliary transport. Apical MTs, which are bound to the BF of BBs, play a crucial role in achieving the positional and orientational orders of BBs. Most importantly, consideration of the symmetry led us to propose that polarity should be maintained in the MT bundle filaments for properly guiding orientation in BBs. This idea is supported by the biased distribution of angles between BB orientation and MT filaments in TEM images (Fig 1D). Although more direct evidence of the presence of polarity in MT bundles is still lacking, this can be investigated and confirmed in future experiments.

We propose a mathematical model for patterning of BBs, taking into consideration the polarity of MT bundles. The model was constructed based on assumptions such as CSK characteristics and volume exclusion among BBs and CSKs. Since the fundamental mechanism of how BBs align and orient is not yet known, the model was not aimed at being precise or quantitative, but rather provided a basic description of the processes for understanding the underlying mechanism following the generic formalism [13–15]. The derived model is distinct from the previous one [3] in incorporating polarity as well as the repulsive interaction between BBs and CSKs, by which it can reproduce both the positional and orientational orders of BBs in multi-ciliated cells. Analysis of the model demonstrated that polarity is crucial not only for determination of BB orientation, but also for achieving alignment pattern of BBs, the latter depending on the nematicity of the CSK determined by polarity. It was also shown that BBs can align even without the condensation effect of CSK concentration, i.e., exclusion of BBs and CSKs was sufficient for the formation of alignment pattern. However, the formation of MT bundles enabled robust formation of the alignment pattern against external disturbance. These simulation results are feasible given the experimental observation that MTs are indispensable for establishing alignment pattern in the apical membrane of multi-ciliated cells where MT bundles segment the aligned array of BBs. Positional order of BBs was disrupted by suppressing MT polymerization with BB clusters being formed, which could also be reproduced by the present model. Taken together, it was demonstrated that incorporation of CSK polarity in the model was necessary and sufficient to reproduce prior experimental observations.

It should be noted that our model still depends on various assumptions and misses some crucial aspects observed in experiments. First, in our model, the effect of PCP was simplified into a fixed boundary condition (i.e., Dirichlet condition) that represents accumulated tyrosinated MTs on the oral side of cellular cortex [21]. As explained before, this accumulation of the cortical MTs itself can be regulated by both PCP proteins and apical MTs. These processes should be incorporated to study a more precise time course of patterning of BBs. Second, in our model, it was assumed that the orientation of BBs merely follows the direction of MTs (process (v) in the model assumption) for which the detailed mechanism was not taken into account. Since BF contains γ-tubulin [9], it will be interesting to explore whether nucleation of the apical MTs at BF has a possible impact on the determination of BB orientation to MTs. In addition, while PCP and apical CSKs are indispensable for the coordination of BBs and ciliary

beating, it was reported that hydrodynamic interaction or direct physical contacts among cilia were part of a positive feedback loop for accurate BB patterning and contributed to the correct directionality in BBs and ciliary beating to enhance mucociliary transport [19,26,32]. It would be interesting to extend our model to understand how the mechanical perturbation from the synchronously beating cilia to the BBs in apical membranes enables smooth BB rearrangement and promotes their alignment pattern.

Finally, we would like to discuss the implication of our study in determining cellular chirality. Presence of polarity in CSKs aligning perpendicular to the PCP axis (oral to lung axis) indicates that the cell can distinguish between left and right directions from the PCP axis (Figs 1A and 5A, blue arrows). From a theoretical point of view, our study provides a possible mechanism for implementing cellular chirality by using PCP and CSK polarity. In the biological system, chirality appears at various levels ranging from molecular [33] and cellular [34] to the tissue level [35]. Cellular chirality is responsible for body formation. For example, cell intrinsic chirality underlies left-right asymmetry of chicken cardiac looping [36,37]. Nodal cilia are well-known examples in which the cilia are tilted in a definite direction and generate the left-ward flow on the nodal tissue surface, which induces the left-right asymmetry of animal body [7,38,39]. Although the mouse trachea studied in the present work has no apparent chiral asymmetry, it is possible that the system can regulate cellular chirality through some mechanisms, which is of potential importance in generating chirality at the tissue level. We believe that our model provides useful insights into exploring the common mechanisms of cellular and tissue chirality.

## Methods

### Mathematical model for BB pattern formation in multi-ciliated cells

Details of the model Eqs (5–7) in the main text are as follows.

$$\partial_t c = \alpha_J \nabla^2(-\alpha_2(c - c_0) + \alpha_4(c - c_0)^3 - D_0(\nabla^2 c + \alpha_Q \nabla \cdot (Q \nabla c)) + 2\lambda_{c\phi} c \sum_i \phi_i) + k_p \frac{c}{c + K_M}$$
$$- k_d c \tag{10}$$

$$\partial_t \boldsymbol{p} = \gamma_P((\beta_c(c) - |\boldsymbol{p}|^2)\boldsymbol{p} + K\nabla^2 \boldsymbol{p} - \alpha_Q D_0 M_c \boldsymbol{p}) \tag{11}$$

$$\partial_t \boldsymbol{X_i} = 2\gamma_{BB} \nu \int \left[ \left( \lambda_{c\phi} c^2 + \lambda_{\phi\phi} (\sum_{j \neq i} \phi_j) \right) \phi_i (1 - \phi_i) \frac{\boldsymbol{X_i} - \boldsymbol{r}}{|\boldsymbol{X_i} - \boldsymbol{r}|} \right] dr + \xi_i \tag{12}$$

Ginzburg-Landau potential $f_c(c) = -(\alpha_2/2)(c - c_0)^2 + (\alpha_4/4)(c - c_0)^4$ has been employed for this model which has two minima at $c^{\pm} = c_0 \pm \sqrt{\alpha_2/\alpha_4}$. The last term in Eq (11) represents the reactive term to align the polarity along the CSK concentration gradient, where $M_c = (\nabla c) \otimes (\nabla c) - (1/2)\text{Tr}[(\nabla c) \otimes (\nabla c)]I$. The reactive term associated with $\beta_c(c)$ was ignored for simplicity in Eq (10). This was verified by interpreting that the emergence of polarity $\boldsymbol{p}$ is governed by active dynamics. In practice, the results are almost independent of the presence of the reactive term. In Eq (3), we adopted $F_{c\phi} = \lambda_{c\phi}\phi_i c^2$ for exclusive volume effect between $i$-th BB and MTs. Alternative symmetrically permissible form is to adopt $F_{c\phi} = \tilde{\lambda}_{c\phi}\phi_i c$, by which we found similar patterns with appropriate choice of $\tilde{\lambda}_{c\phi}$ and $\lambda_{\phi\phi}$.

The dynamics of CSK and BBs was numerically simulated by integrating the differential Eqs (8 and 10–12). The non-dimensionalized equations of CSK defined on a rectangular domain $L_x \times L_y$ were calculated. The time integration is done by the fourth-order finite difference scheme with time step $\Delta t = 1.0 \times 10^{-3}$. Periodic boundary condition was employed unless

stated otherwise. The system size was chosen as $L_x = L_y = 64$, for which the domain is discretized by a square grid with $\Delta x = 1.0$ spacing. The number of BBs is $N = 80$ for most of the simulations. Numerical simulations were performed with system size $L_x = L_y = 32$ and $N = 20$ for producing the phase diagrams in Figs 3 and 4. Four independent simulations were performed for each set of parameters to check the dependency on initial conditions. Initial conditions were set as follows. The position and orientation of BBs, $X_i$ and $\theta_i$, were random, the polarity of CSK $p$ was set at $|p| = 0.1$ with random direction, and the concentration of CSK is set at $c = 0.2$ with small value of additional noise.

The values of the parameters used in the simulations are summarized in Table 2. Since exact kinetic values were not determined experimentally, we determined them so as to produce experimental patterns, using the following rationale. The units of length and time were set to be equal to 0.04 $\mu m$ and 5 min, respectively. Radius of a BB was chosen as $R = 2.5$ corresponding to 0.1 $\mu m$ [3,11]. System size was $L_x = L_y = 64 \times 64$ corresponding to $2.56 \times 2.56\ \mu m^2$, approximately quarter of the apical area of single multi-ciliated cells. $N = 80$ since ~300 BBs exist per cell. Typical speed of BBs was about 0.02–0.03$\mu$m/ 5min [3]. Hence, $\gamma_{BB}$ was chosen to satisfy $dX_i/dt \sim \gamma_{BB}R = 0.03\mu$m/ 5min. The concentration of CSKs was not known. Therefore, we normalized it and used the dimensionless concentration for $c$ by setting parameters to ensure $c^- = 0$ and $c^+$ to be almost unity. Amplitude of polarity $|p|$ was also chosen to be almost unity. Another prerequisite for $c$, modeled using Ginzburg-Landau potential combined with the reaction term [40], was that the system must show bundling. Reaction rate $k_d$ should be smaller than the critical value evaluated by linear stability analysis, $k_d^c = \alpha_J \alpha_2^2(1 + K_M/c_0)/(4D_0) = 1.05$ (see S1 Fig). Typically, we used $k_d^{-1} = 3.16$ which corresponds to ~15 min in reality. This is comparable with the observed time scale of MT bundle dynamics in vitro [41]. In addition, we set $\alpha_J = \gamma_P K$ following the model for actin filament in [15]. With these choice of parameters, the pattern formation of BBs took 600~1200 time in the simulations (Figs 2 and 5), corresponding to 2~4 days. This is consistent with actual developing time and experimental observation in cultured tracheal cells [3]. We should note that the parameter values themselves, such as equilibrium concentration of MTs ($k_p$ and $k_d$), can be gradually changed during the developmental stage, which could govern the time scale of patterning ultimately.

We set $k_d = 1.0 \times 10^{-2}$ to obtain the results in Fig 4D, where polarity is ignored and the dynamics of CSK and BBs are simulated by using Eq (12) and

$$\partial_t c = \alpha_J \nabla^2 \left( -D_0 \nabla^2 c + 2\lambda_{c\phi} c \sum_i \phi_i \right) + k_p \frac{c}{c + K_M} - k_d c. \tag{13}$$

We set the system size as $N = 30$, $L_x = 30$, and $L_y = 45$ to simulate the model coupled with PCP information shown in Fig 5. As briefly mentioned in the main text, Neumann boundary condition was initially employed, followed by time step $t = 66.7$ h. The boundary condition of the oral side (top boundary in Fig 5C) was changed to Dirichlet where CSK concentration was fixed at $c = c_H$ to mimic the experimental observation that the distribution of MTs is dense in the opposite side of the cell boundary where PCP protein Vangl1 is accumulated (i.e., lung side) [3,21]. $c_H$ was chosen as $c_H = c^+ = c_0 + \sqrt{\alpha_2/\alpha_4}$. Polymerization rate $k_p$ was set to be an increasing function of time as $k_p/k_d = 0.6 + 0.1 \tanh(1.0 \times 10^{-3}t - 0.5)$ (Fig 5B, blue line), since MTs and intermediate filaments increase during maturation of cells [12].

## Model without BBs

It is useful to summarize the case where BBs are absent, to understand the behavior of the model. The model is composed of phase-separation dynamics with first order chemical

reaction [42] and polarity dynamics [13]. The phase separation is driven by Landau-Ginzaburg potential $f_c(c)$. We have numerically solved Eqs (10 and 11) without the terms representing interaction with the BBs.

As shown in S1 Fig, the pattern of CSK concentration is dependent on $\alpha_Q$ and the polymerization/depolymerization parameter $k_d$ [42]. For lower chemical reaction rate (i.e., smaller $k_d$), the CSK domain concentration is larger. CSK concentration shows stripe pattern in the middle range of $k_d$. The range of aligned stripe is wider when the coupling constant with polarity, $\alpha_Q$, is larger, while it is very narrow at $\alpha_Q = 0$. In addition, the stripe pattern is less sharp for smaller $\alpha_Q$. The CSK concentration becomes uniform for larger value of $k_d$. In summary, the model shows aligned robust bundles of CSKs for proper value of $k_d$ and higher value of $\alpha_Q$. The model without BBs satisfies the minimal prerequisite to express the bundling of CSKs in our model.

## Supporting information

**S1 Fig. Patterns of CSKs without BBs.** The patterns are dependent on $\alpha_Q$ and $k_d$; Striped pattern appears for finite value of $\alpha_Q$ (striped patterns which are not sharp may appear at $\alpha_Q = 0$ in a narrow parameter region). CSKs concentration becomes uniform at very high values of polymerization/depolymerization rate $k_d$. Bars, 0.5 μm.
(TIF)

**S2 Fig. BB positional order depends on the elasticity parameter, *K*.** Color indicates the straightness of BB quantified by $\langle -\cos\psi_i \rangle$. Inset figures show typical pattern of BBs. Gray dashed line ($K = 3.0$) corresponds to the line connecting between symbol A and PA in Fig 3. Bars, 0.5 μm.
(TIF)

**S3 Fig. Dependence of BB pattern on the form of condensation term $f_c(c)$.** The form of $f_c$ is expressed as $f_c(c) = \lambda_W(c-c_0-W)^2(c-c_0+W)^2$. Left: phase diagram against $\lambda_W$ and $W$. Color indicates the straightness of BB ($\langle -\cos\psi_i \rangle$). Right: Four representative patterns are shown (i-iv) with corresponding black squares in the left panel. Bars, 0.5 μm.
(TIF)

**S1 Video. Model simulation of BBs corresponding to Fig 2B.** BBs reach "alignment" pattern. Bars, 0.5 μm.
(MP4)

**S2 Video. Model simulation of BBs corresponding to Fig 2C.** BBs reach "partial alignment" pattern. Bars, 0.5 μm.
(MP4)

**S3 Video. Model simulation of BBs without the condensation term, corresponding to Fig 4A.** Bars, 0.5 μm.
(MP4)

**S4 Video. Model simulation for nocodazole treated cells, corresponding to Fig 4D.** Bars, 0.5 μm.
(MP4)

**S5 Video. Model simulation of BBs aligned by PCP, corresponding to Fig 5C.** Bars, 0.5 μm.
(MP4)

## Acknowledgments

We thank Prof. Sachiko Tsukita and her laboratory members for helpful discussions, especially A. Tamura, T. Yano, S. Konishi, D. Taniguchi, E. Herawati, S. Nakayama, H. Kanoh, for helpful discussions and showing us their original data. We also thank Dr. D. Taniguchi for his useful advices.

## Author Contributions

**Conceptualization:** Shuji Ishihara.

**Data curation:** Toshinori Namba.

**Formal analysis:** Toshinori Namba.

**Funding acquisition:** Shuji Ishihara.

**Investigation:** Toshinori Namba.

**Methodology:** Toshinori Namba, Shuji Ishihara.

**Project administration:** Shuji Ishihara.

**Supervision:** Shuji Ishihara.

**Validation:** Toshinori Namba.

**Visualization:** Toshinori Namba.

**Writing – original draft:** Toshinori Namba, Shuji Ishihara.

**Writing – review & editing:** Toshinori Namba, Shuji Ishihara.

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
