## [Decision Letter · Decision Letter 0]

7 Sep 2019

Dear Dr Ishihara,

Thank you very much for submitting your manuscript 'Cytoskeleton polarity is essential in determining direction in basal bodies of multi-ciliated cells' for review by PLOS Computational Biology. Your manuscript has been fully evaluated by the PLOS Computational Biology editorial team and in this case also by independent peer reviewers. The reviewers appreciated the attention to an important problem, but raised some substantial concerns about the manuscript as it currently stands.

In particular, the model presented is a next step over your previous model presented in Ref. [3], in that the new model considers the polarity of the CSKs. Your new work shows that if the polarity of the CSKs, i.c. microtubules (MTs) is considered, this can explain how the BB arrays become orientationally ordered. However, reviewer 1 notes that such polarity/parallel ordering is likely not found - the MTs are ordered antiparallel with a slight bias at best. The reviewer argues that, in cases where such slight MT orientation is found, it is likely to be the result, not the cause of BB orientation. Also if cilia motion is disrupted, thus disrupting fluid flow, the orientation of the BBs is lost; although this key role of fluid flow is mentioned in the discussion, the model does not provide an explanation for this discrepancy with the experimental observations.

While your manuscript cannot be accepted in its present form, we are willing to consider a revised version in which these issues and the other issues raised by the reviewers have been adequately addressed. We cannot, of course, promise publication at that time.

Sincerely,

Roeland M.H. Merks, Ph.D

Associate Editor

PLOS Computational Biology

Mark Alber

Deputy Editor

PLOS Computational Biology

[LINK]

Reviewer's Responses to Questions

**Comments to the Authors:**

Reviewer #1: The manuscript by Namba and Ishihara deals with the fascinating question of how cilia in a multiciliary array become coordinately polarized. The authors provide a mathematical model for how this is achieved that relies heavily on the interactions between basal bodies and the microtubule cytoskeleton. The authors do an admirable job of summing up the relevant literature and there is certainly some merit to their model. However, I have several serious concerns that warrant consideration.

1. MT bundles can be parallel or anti parallel. The information in the literature would suggests that these MT bundles are mixed or anti parallel, but the current equations if I understood them correctly rely on parallel bundles all going in the same direction. This is not likely to be the case. At the very best there will be a slight bias in the directionality of the MTs. I think this mixed array needs to feature more heavily into any equations.

2. Similar to above the authors state “Although more direct evidence of the presence of polarity in MT bundles is still lacking, this can be investigated and confirmed in future experiments.” Multiple groups have tried to do this in multiple systems. There is simply not a strong directional bias to the MT orientation before BB polarization. Vladar et al. 2013, Boutin et al 2014, Kim et al 2018 all show some polarized accumulation of MTs in MCCs but this comes after BB polarization and likely reflects the bias in orientation of MTs nucleating from the BF that are all pointing in the same direction. If oriented MTs were defining the polarity for BB then it would come first, which does not seem to be the case. Additionally, all of the mentioned data have polarized subset of MTs parallel with the axis of orientation (and primarily at the cortext) whereas this model is based on the pool of MTs interacting with BBs which must be more perpendicular to the axis of orientation.

3. The authors state “However, this binding between BF and MTs allows BBs to associate with either of the two neighboring MT bundles and hence they cannot distinguish between the two opposite directions (Fig 1C (i)). Therefore, how BBs can determine their definite direction remains a mystery.” If I understand things correctly, the authors are treating the MTs and the BB independently. Figure 1C suggests that the BF are interacting with the MTs, which is somewhat consistent with the EM literature. However, what I think is missing from this discussion is where the MTs are originating. I think there is strong indications in the literature that the MTs are in fact at least in part nucleating from the BF. Therefore it is not really appropriate to consider these things completely independent of one another.

4. Related to above all MTs in this manuscript are created equal which is almost certainly a critical oversimplification. The evidence from Vladar et al. would suggest that there is a distinct pool of MTs labeled with tyrosinated tubulin that interact with cell cortex in a polarized manner. However the majority of MTs are not tyrosynated in these cells.

5. Finally and perhaps most concerning is the lack of integration of the key information that hydrodynamic forces are critical to the polarization of these BBs. While the authors mention this in their discussion it is completely lacking from the model. We know from multiple systems that if cilia motility is disrupted (with the rest of the MT cytoskeleton intact) then BBs will not orient. This tells us that the MT BB interactions alone are not sufficient to drive orientation. While I appreciate that this is a complicated question and the connection of MTs and BBs is an important aspect, a model that does not explain the basic experimental evidence needs improvement.

Minor:

“In particular, the attachment of BF to MTs might be important for the determination of BB direction, but its mechanistic details have not been identified as yet.” I more or less agree with this statement and also feel that in general the authors have done a thorough job of referencing the literature, however I think referencing the Turk et al paper on gamma tubulin at the basal foot might be appropriate as this is probably the closest we have seen to mechanism for nucleating MTs from the BF.

Reviewer #2: Uploaded attachment

**Have all data underlying the figures and results presented in the manuscript been provided?**

Reviewer #1: Yes

Reviewer #2: Yes

PLOS authors have the option to publish the peer review history of their article (what does this mean?). If published, this will include your full peer review and any attached files.

Reviewer #1: No

Reviewer #2: No

---

## [Decision Letter · Decision Letter 1]

9 Jan 2020

Dear Dr Ishihara,

We are pleased to inform you that your manuscript 'Cytoskeleton polarity is essential in determining orientational order in basal bodies of multi-ciliated cells' has been provisionally accepted for publication in PLOS Computational Biology.

In the meantime, please log into Editorial Manager at https://www.editorialmanager.com/pcompbiol/, click the "Update My Information" link at the top of the page, and update your user information to ensure an efficient production and billing process.

One of the goals of PLOS is to make science accessible to educators and the public. PLOS staff issue occasional press releases and make early versions of PLOS Computational Biology articles available to science writers and journalists. PLOS staff also collaborate with Communication and Public Information Offices and would be happy to work with the relevant people at your institution or funding agency. If your institution or funding agency is interested in promoting your findings, please ask them to coordinate their releases with PLOS (contact ploscompbiol@plos.org).

Thank you again for supporting Open Access publishing. We look forward to publishing your paper in PLOS Computational Biology.

Sincerely,

Roeland M.H. Merks, Ph.D

Associate Editor

PLOS Computational Biology

Mark Alber

Deputy Editor

PLOS Computational Biology

Reviewer's Responses to Questions

**Comments to the Authors:**

Reviewer #1: In general I think modeling studies are really important and so I like to be supportive. However, I also get very frustrated by them as they are always over simplifications. I think this study is reasonable for what it is but they have not been able to deal with my primary concern that their model does not incorporate what the field knows is a critical aspect, namely hydrodynamic forces. I appreciate that this would be challenging and they certainly mention it and propose to add it in future. But as a researcher in the field I am not sure this work changes anything about the way I think about this question. The discussion on chirality is interesting but seems out of place in the current work. Overall, I think this work does a good job of addressing some of the data in the field and thus it does have merit, and it is clear that it is hard to isolate the importance of individual factors when we know that these are part of a larger combined effort. I think the authors should be commended on their effort and I think it is a worthwhile contribution that may be the foundation for future more inclusive models. As such I think it has value to the field and I am supportive of publication.

Reviewer #2: The manuscript is considerably improved in terms of readability. I enjoyed reading it a second time. The manuscript is well and very carefully written, including multiple justifications and considerations that can serve as starting work for future studies. The introduction and discussions have also been enriched. I'm happy to recommend it for publication.

**Have all data underlying the figures and results presented in the manuscript been provided?**

Reviewer #1: Yes

Reviewer #2: Yes

PLOS authors have the option to publish the peer review history of their article (what does this mean?). If published, this will include your full peer review and any attached files.

Reviewer #1: No

Reviewer #2: No

---

## [Editor Report · Acceptance letter]

13 Feb 2020

PCOMPBIOL-D-19-01166R1 

Cytoskeleton polarity is essential in determining orientational order in basal bodies of multi-ciliated cells

Dear Dr Ishihara,

I am pleased to inform you that your manuscript has been formally accepted for publication in PLOS Computational Biology. Your manuscript is now with our production department and you will be notified of the publication date in due course.

With kind regards,

Sarah Hammond
